*Resource*

# Phospho-RNA-seq: a modified small RNA-seq method that reveals circulating mRNA and lncRNA fragments as potential biomarkers in human plasma

Maria D Giraldez[1,2,3,†] (iD), Ryan M Spengler[1,†] (iD), Alton Etheridge[4], Annika J Goicochea[1], Missy Tuck[1], Sung Won Choi[5], David J Galas[4] (iD) & Muneesh Tewari[1,6,7,8,*] (iD)

## Abstract

Extracellular RNAs (exRNAs) in biofluids have attracted great interest as potential biomarkers. Although extracellular microRNAs in blood plasma are extensively characterized, extracellular messenger RNA (mRNA) and long non-coding RNA (lncRNA) studies are limited. We report that plasma contains fragmented mRNAs and lncRNAs that are missed by standard small RNA-seq protocols due to lack of 5′ phosphate or presence of 3′ phosphate. These fragments were revealed using a modified protocol ("phospho-RNA-seq") incorporating RNA treatment with T4-polynucleotide kinase, which we compared with standard small RNA-seq for sequencing synthetic RNAs with varied 5′ and 3′ ends, as well as human plasma exRNA. Analyzing phospho-RNA-seq data using a custom, high-stringency bioinformatic pipeline, we identified mRNA/lncRNA transcriptome fingerprints in plasma, including tissue-specific gene sets. In a longitudinal study of hematopoietic stem cell transplant patients, bone marrow- and liver-enriched exRNA genes were tracked with bone marrow recovery and liver injury, respectively, providing proof-of-concept validation as a biomarker approach. By enabling access to an unexplored realm of mRNA and lncRNA fragments, phospho-RNA-seq opens up new possibilities for plasma transcriptomic biomarker development.

**Keywords** cell-free RNA; extracellular RNA; liquid biopsy; RNA-seq
**Subject Categories** RNA Biology
**The EMBO Journal (2019) 38: e101695**

## Introduction

In recent years, the discovery of a variety of extracellular RNA (exRNA) molecules present in the human bloodstream and other biofluids has been of great interest given their potential value as minimally invasive biomarkers for a wide range of diseases (Freedman *et al*, 2016; Yuan *et al*, 2016; Godoy *et al*, 2018; Max *et al*, 2018). To date, characterization of exRNAs in blood has mostly focused on microRNAs, which have been shown to be exceptionally stable in plasma (i.e., the acellular portion of blood) by virtue of being protected in complexes with Argonaute proteins and extracellular vesicles (Hunter *et al*, 2008; Arroyo *et al*, 2011). However, microRNAs represent a small fraction of the human transcriptome and only a small minority of microRNAs show exquisite tissue or disease specificity (Ludwig *et al*, 2016). The degree to which the more predominant components of the transcriptome, notably mRNAs and lncRNAs, are represented in blood as exRNA is not clearly established. Yet, mRNAs and lncRNAs are highly appealing from the standpoint of biomarkers for monitoring health and disease due to their multiple established tissue- and disease-specific gene expression signatures (Perou *et al*, 2000; Potti *et al*, 2006; Chen *et al*, 2007; Ben-Porath *et al*, 2008; Liu *et al*, 2008; Iyer *et al*, 2015).

RNA-seq has transformed transcriptome characterization in a wide range of biological contexts (Mortazavi *et al*, 2008; Wang *et al*, 2009) including its application to analyze exRNA in body fluids (Adiconis *et al*, 2013; Giraldez *et al*, 2018). These efforts have begun to elucidate the complex composition of exRNA in blood (Freedman *et al*, 2016; Yeri *et al*, 2017; Godoy *et al*, 2018; Max *et al*, 2018). There have been indications of extracellular mRNA and lncRNA in some studies of plasma, but results have been

1    Department of Internal Medicine, Hematology/Oncology Division, University of Michigan, Ann Arbor, MI, USA
2    Institute of Biomedicine of Seville (IBiS), Seville, Spain
3    Unit of Digestive Diseases, Virgen del Rocio University Hospital, Seville, Spain
4    Pacific Northwest Research Institute, Seattle, WA, USA
5    Department of Pediatrics, Hematology/Oncology Division, University of Michigan, Ann Arbor, MI, USA
6    Center for Computational Medicine and Bioinformatics, University of Michigan, Ann Arbor, MI, USA
7    Department of Biomedical Engineering, University of Michigan, Ann Arbor, MI, USA
8    Biointerfaces Institute, University of Michigan, Ann Arbor, MI, USA
     *Corresponding author. Tel: +1 734 764 2616; E-mail: mtewari@med.umich.edu
     †These authors contributed equally to this work

inconsistent, with some profiling studies reporting a variable percentage of them and others not even reporting their presence (Huang *et al*, 2013; Koh *et al*, 2014; Freedman *et al*, 2016; Yuan *et al*, 2016; Danielson *et al*, 2017; Yeri *et al*, 2017; Godoy *et al*, 2018; Max *et al*, 2018). Moreover, these profiling studies have used a variety of methods to evaluate exRNA (e.g., microarrays and different methodologies for RNA-seq) which, not surprisingly, contributes to the variation in findings.

We hypothesized that given the high concentration of RNases in the human bloodstream (Kamm & Smith, 1972), mRNAs and lncRNAs, if truly present in blood plasma at all, may not exist in full-length form, but rather as small fragments. Furthermore, we hypothesized that standard ligation-based small RNA-seq methods might not detect such fragments because they are designed to capture microRNAs (Hafner *et al*, 2008), which by virtue of being products of RNase III class enzymes (e.g., Dicer) consistently present 5′ monophosphate (5′ P) and 3′ hydroxyl (3′ OH) ends (Lee *et al*, 2003). In contrast, the 5′ and 3′ ends of RNA cleavage products generated by other ribonucleases vary substantially, which might prevent efficient adapter ligation with typical small RNA-seq methods. For example, abundant RNases in human blood circulation, such as those belonging to the ribonuclease A superfamily (Lu *et al*, 2018), degrade RNA dinucleotide bonds, leaving a 5′ OH and 3′ P product (Cuchillo *et al*, 2011). Therefore, we reasoned that in order to sequence a broader space of exRNAs beyond microRNAs, it would be essential to develop modifications to small RNA-seq protocols that can enable capture of RNA fragments that may have these alternate 5′ and 3′ phosphorylation states.

Here, we modified the standard small RNA-seq approach by incorporating both an upfront 5′ RNA phosphorylation/3′ dephosphorylation step using T4 polynucleotide kinase (referred to here subsequently as "PNK") and a custom, high-stringency bioinformatic data analysis pipeline to analyze non-microRNA small RNA fragments. This approach, which we refer to as "phospho-RNA-seq", revealed a large, untapped space of mRNAs and lncRNA fragments present in plasma. These fragments comprised tissue-specific signatures that reflected biological processes of bone marrow reconstitution and acute liver injury in hematopoietic stem cell transplant (HSCT) patients. We propose that this approach opens up new opportunities for disease biomarker discovery through transcriptomic analysis of exRNA fragments in circulation.

## Results

### Synthetic RNA-based technical validation of a phospho-RNA-seq protocol for recovering short mRNA and lncRNA fragments with ends lacking a 5′ P and/or possessing a 3′ P

To evaluate the performance of both standard and phospho-RNA-seq methods for recovering short oligonucleotides with varying end modifications likely to be found in human biofluids, we designed a synthetic reference pool comprising 476 ribonucleotides of different length (from 15 to 90 nt) and sequence (Table EV1). More specifically, our pool includes 286 human microRNAs, 8 plant microRNAs, 164 fragments of mRNA and lncRNAs ranging from 15 to 90 nt and including different end modifications (5′ P + 3′ OH, 5′ OH + 3′ P and 5′ OH + 3′ OH), and 18 artificial microRNA sequences as

controls. As depicted in Fig 1A, we prepared small RNA libraries using this pool as input and following two different strategies: (i) standard ligation-based methodology (i.e., TruSeq small RNA protocol) and (ii) our modified phospho-RNA-seq approach (i.e., RNA pretreatment with PNK which phosphorylates 5′ hydroxyl groups and removes 3′ phosphoryl groups from oligonucleotides, followed by standard small RNA library preparation methodology). Libraries were multiplexed, sequenced on a NextSeq platform, and analyzed as described in Materials and Methods.

As shown in Fig 1B, both strategies were able to recover the majority of sequences with 5′ P and 3′ OH in our pool, most of which are human microRNAs. In contrast, the phospho-RNA-seq approach recovered sequences that either lacked a 5′ P or had a 3′ P, which were largely undetectable by the standard methodology (Fig 1B–D). We confirmed that these results were not due to differences in sequencing depth, as the untreated library generated 4.1 million aligned reads, compared with only 2.9 million in the PNK-treated library. These results confirmed that standard ligation-based small RNA protocols are poorly suited for capturing non-microRNA species lacking a 5′ P and, especially, those presenting a 3′ P.

### Phospho-RNA-seq combined with a high-stringency bioinformatic pipeline enables reliable detection of mRNA/lncRNA fragments in human plasma

After validating the efficiency of our phospho-RNA-seq strategy for capturing mRNA and lncRNA fragments with a variety of end modifications in a setting where the ground truth is known (i.e., a synthetic pool of RNA), we aimed to design and test a pipeline that could enable reliable evaluation of mRNA and lncRNA fragments in real plasma samples, where the exRNA composition is unknown and the risk of false-positive calling is higher. To this end, we obtained platelet-poor plasma from five healthy control individuals (demographic features are shown in Appendix Table S1), prepared triplicate libraries for each individual using both standard small RNA-seq methodology (TruSeq kit) and phospho-RNA-seq, and performed multiplexed sequencing on a HiSeq platform (Fig 2A).

Initial attempts to characterize mRNA and lncRNA fragments directly from adapter-trimmed and length-filtered reads revealed that non-mRNA/lncRNA sequences, including fragments from various endogenous non-coding RNAs (defined here as rRNAs, tRNAs, microRNAs, and other non-coding RNAs but excluding lncRNAs, as derived from GENCODE and described in detail in Materials and Methods) and repetitive elements, were leading to false-positive detection and over-estimation of mRNA and lncRNA fragment abundances. To uncover relevant biological signal derived from mRNAs and lncRNAs, we developed a custom pipeline (Fig 2B) that employs multiple distinct filtering steps aimed at quantifying and removing potential sources of false signal, to enable the reliable detection of short mRNA and lncRNA fragments. Stage 1 of the pipeline involves trimming adapters, removing low-quality bases, and eliminating reads shorter than 15 nucleotides (see Materials and Methods for additional criteria). Next, we adapted the sRNAnalyzer pipeline (Wu *et al*, 2017) to quantify and remove reads aligning to any one of several sequence libraries containing exogenous RNAs (bacterial, fungal, and viral), various endogenous non-coding RNA sequences, and other possible contaminants (transposons, repetitive elements, and UniVec contaminants; Stage 2). Reads with no valid

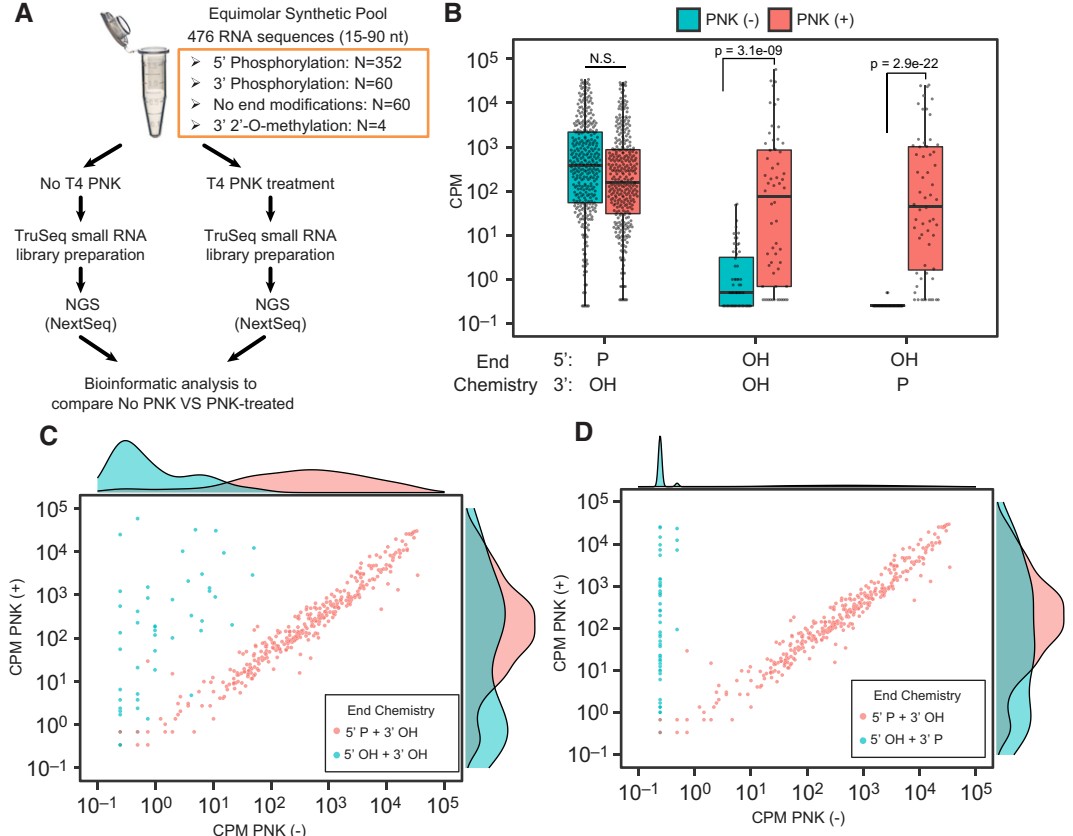

**Figure 1.  A modified protocol overcomes the low efficiency of standard ligation-based small RNA library preparation methods for cloning short RNA sequences lacking a 5′ P or possessing a 3′ P.**

A     Schema of experimental design**.**
B     Boxplots summarize the mean counts per million (CPM) observed for sequences contained in the synthetic equimolar pool (*y-axis*, log$_{10}$ scaled) presenting different end modifications (*x-axis*), as measured from libraries prepared using a standard ligation-based small RNA protocol (PNK (−)) and the phospho-RNA-seq strategy (PNK (+)). Boxes represent the mean ± interquartile range (IQR), and whiskers represent 1$^{st}$/3$^{rd}$ quartile 1.5 * IQR. Boxplots summarize mean CPM values for $n$ = 352 (5′ P + 3′ OH sequences), $n$ = 60 (5′ OH + 3′ OH sequences), and $n$ = 60 (5′ OH + 3′ P sequences). Significant Bonferroni-adjusted *P*-values are shown from one-tailed Wilcoxon rank-sum tests for differences in abundance between PNK (+) and PNK (−), for sequences with the end chemistries shown (alternative hypothesis PNK+ > PNK−).
C, D   Scatter plots showing the read distribution (CPM) observed for sequences of the synthetic equimolar pool with 5′ P (red dots) and (C) without end modifications (teal dots) or (D) with 3′ P (teal dots) as measured from libraries prepared using a standard ligation-based small RNA protocol (*x*-axis) and the phospho-RNA-seq strategy (*y*-axis). Marginal density plots are included as a summary of the data.

alignments to these sequence libraries in Stage 2 are then aligned to the human genome. In Stage 3, genomic read alignments are filtered if found to have any overlap with RepeatMasker (UCSC) and various endogenous non-coding annotation coordinates. This additional coordinate-based filtering step catches reads that were missed by the sRNAnalyzer workflow (see Materials and Methods for additional details).

As shown in Fig 2C, without any filtering of non-coding RNA and repeat-mapping reads, thousands of mRNA and lncRNA genes were falsely detected or detected at artificially high levels due to a preponderance of reads aligning to transcript-embedded fragments from various endogenous non-coding RNA or repetitive element sequences. The alignment to sequence databases in Stage 2 and the coordinate-based filtering in Stage 3 provided a step-wise removal of false positives from these endogenous sources (Fig 2C). Accordingly, the percentage of reads uniquely mapped to mRNA and lncRNA exons also increases through the sequential filtering stages

of our pipeline (Fig 2D). Therefore, our analysis demonstrates the importance of stringent filtering of repetitive sequences and certain non-coding RNA sequences, as failure to do so resulted in false-positive detection of many mRNA/lncRNA transcripts. It is also worth noting that sequences from libraries prepared with phospho-RNA-seq mapped more frequently to mRNA and lncRNA exons than those prepared using standard small RNA-seq, with the former showing a 10-fold increase in mRNA/lncRNA exonic reads on average (Fig 2E).

**Standard ligation-based small RNA-seq pipelines are prone to false-positive calling of mRNA/lncRNA fragments in human plasma**

To evaluate how reliable standard small RNA-seq pipelines are for calling short mRNA and lncRNA fragments, we processed the

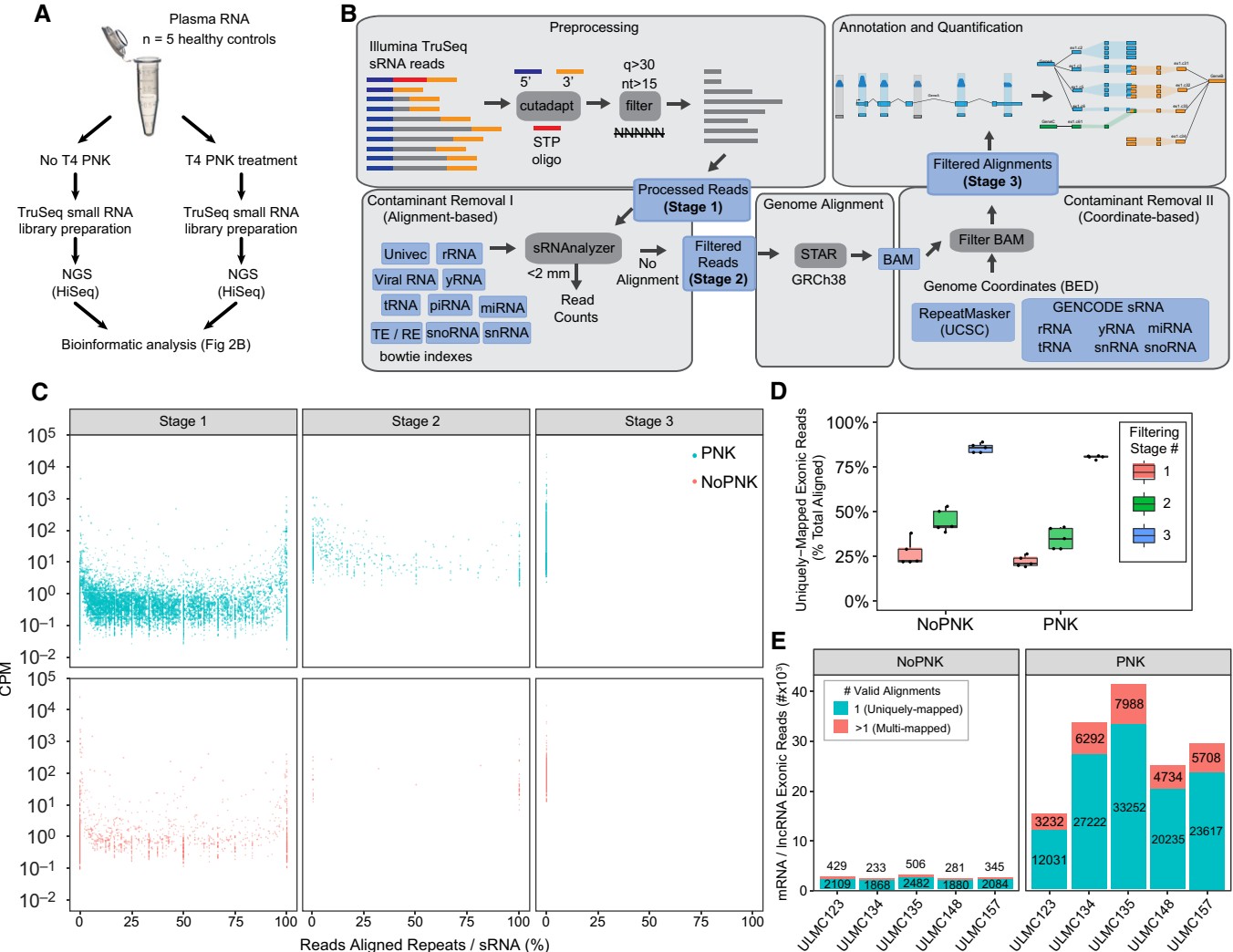

**Figure 2. Phospho-RNA-seq combined with stringent contaminant sequence filtering reduces false-positive mRNA/lncRNA fragments.**

A   Schema of experimental design.

B   Schema of bioinformatic analysis pipeline.

C   Scatter plot showing the percentage of reads aligned to repeats and small RNAs (x-axis) for the each filtering stage of our custom pipeline. Dots represent the mean CPM calculated for each gene across the five healthy control individuals.

D   Boxplots show the fraction of genome alignments that are unambiguously aligned to mRNA and lncRNA exons, shown as the percent total reads aligned at each filtering stage. The points represent and the boxplots summarize the percentages calculated from combining alignments from three technical replicates for each of the (n = 5) healthy individuals. Boxes represent the mean $\pm$ IQR, and whiskers represent $1^{st}/3^{rd}$ quartile 1.5*IQR.

E   Barplots show the number of uniquely mapped (teal) and multi-mapped (red) mRNA and lncRNA exonic reads remaining after the final filtering stage (Stage 3). Counts are plotted for each of the five healthy control individuals (x-axis) both in untreated (left panel) and in PNK-treated (right panel) samples. The read count values are shown along with the corresponding barplots.

plasma exRNA sequencing data from a healthy individual through exceRpt, a pipeline specifically designed for the analysis of exRNA small RNA-seq data that uses its own alignment and quantification engine to map and quantify a range of RNAs including mRNA and lncRNA (see Materials and Methods). We then selected the 50 most abundant mRNA transcripts called by the exceRpt pipeline for evaluation through each stage of our custom high-stringency pipeline. As in our own pipeline, exceRpt first aligns adapter-trimmed reads to several small RNA databases for quantification, and only reads with no valid small RNA alignments are subsequently aligned to the

human genome and used for mRNA and lncRNA quantification. However, tracking the reads corresponding to the top 50 most abundant exceRpt mRNA transcripts through our own high-stringency pipeline (Fig 3A) showed that although all 50 transcripts were detectable at Stage 1, most of them were filtered out or significantly reduced in relative expression by subsequent filtering steps in Stages 2 and 3. We confirmed that a high proportion of them corresponded to fragments from various endogenous non-coding RNA species or repeat-mapping reads that were, therefore, ultimately filtered out in our pipeline (Fig 3B). Interestingly, for the libraries prepared using

phospho-RNA-seq, only 10 of the top 50 exceRpt transcripts were filtered out when analyzed through our highly stringent pipeline, as compared to 35 with standard small RNA-seq. These results demonstrate that standard small RNA-seq pipelines, even thoughtfully designed ones like the exceRpt pipeline, which seek to map and remove some small RNA and repetitive sequence species prior to human genome alignment, are prone to false-positive calling of mRNA and lncRNA fragments, thus limiting reliable identification of these exRNA species in plasma samples.

## Assessment of short mRNA/lncRNA fragments in human plasma using phospho-RNA-seq and our custom, high-stringency bioinformatic pipeline

After having validated that phospho-RNA-seq combined with a custom, high-stringency bioinformatic pipeline enables reliable identification of short mRNA and lncRNA fragments in plasma samples, we sought to assess the abundance and features of these exRNA species in human plasma. In order to further substantiate the validity of these cell-free mRNA fragments, we assessed the relative enrichment of reads aligning in the sense versus antisense orientation. The ligation-based library preparation protocol that we used ensures that the majority of reads are "stranded"—that is, they align in the same orientation as the transcript of origin. Spurious alignments from exogenous RNAs, repetitive sequences, DNA contaminants, or other noise introduced by sequencing artifacts is expected to be distributed more randomly and would result in a more equal distribution of sense/antisense alignments. Thus, as a quality check, we confirmed that the exonic alignments of our plasma exRNA sequence reads were enriched for the sense orientation, relative to antisense, for mRNAs and lncRNAs (Fig 4A). The degree of enrichment for the sense orientation of lncRNAs was lower than for mRNAs, but this may be because the lncRNA database we used includes a diversity of lncRNA types (see Materials and Methods), including those overlapping mRNA transcripts on the opposite strand. Sense strand preference was less evident for reads aligning to introns or promoter regions of mRNA or lncRNA genes, consistent with the expectation that extracellular mRNA and lncRNA fragments result from fragmentation of mature, processed transcripts (Fig 4A). We, therefore, focused our analysis of plasma exRNA on reads aligning to exons of mRNA and lncRNA genes.

We found that our strategy is able to uncover thousands of mRNAs and lncRNAs present in physiological conditions in plasma from healthy individuals (n = 5 healthy controls; Table EV2). We evaluated the read distribution of the mRNA and lncRNA fragments identified in human plasma with our approach (Fig 4B) and found that they are, on average, fairly short (i.e., 20–25 nt range predominantly). However, it is worth mentioning that when we focused our analysis on the top 100 expressed mRNA and lncRNAs or on those mRNA and lncRNA expressed in all the samples, they tended to be slightly longer than the overall population of mRNA/lncRNA fragments, suggesting that longer read lengths are more frequently associated with more abundant and consistently detected genes (Fig 4B and C).

Among the mRNA and lncRNA fragments we found in healthy individuals were these: (i) red blood cell-derived transcripts including several types of hemoglobin transcripts (e.g., HBA1, HBA2, and HBB); (ii) platelet-derived transcripts such as platelet-derived

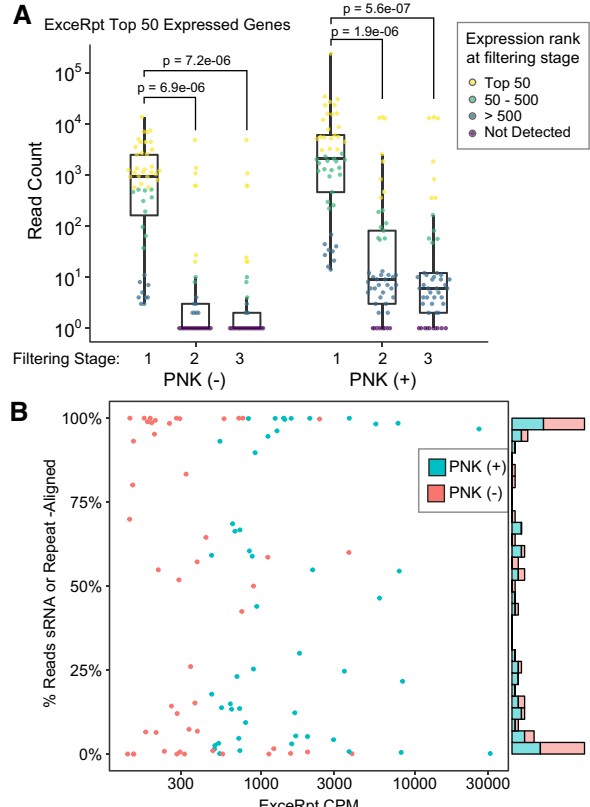

**Figure 3. Re-evaluation of top transcripts called by a standard small RNA-seq analysis pipeline using our custom high-stringency analysis pipeline.**

A The exceRpt exRNA-seq pipeline was used to analyze plasma RNA from a healthy control (ULMC135), and the 50 most highly expressed protein-coding mRNAs were quantified using our pipeline. Boxplots summarize the read counts measured when processed through our repeat filtering stages. Results from both PNK-treated and untreated samples (x-axis) are shown. Gene abundances are shown as $\log_{10}$ read counts + 1 (y-axis). Individual points are color-coded by the rank of the gene expression observed at the stage indicated (rank 1 = highest expressed). Boxes represent the mean ± IQR, and whiskers represent 1st/3rd quartile 1.5*IQR. Bonferroni-adjusted P-values are shown from Wilcoxon rank-sum tests comparing the gene expression ranks in filtering Stage 1 versus Stage 2 or Stage 3.
B Scatter plot shows the CPM values reported by the exceRpt pipeline for the 50 most highly expressed mRNA or lncRNA genes (x-axis), versus the percentage that we found to overlap RepeatMasker or sRNA annotations (y-axis). Values are plotted for ULMC135 + PNK (teal) and − PNK (red).

growth factors (e.g., PPBP); (iii) ubiquitous, highly expressed transcripts such as ferritin chains (i.e., FTH1 and FTL), mitoferrin-1 (i.e., SLC25A37), conventional non-muscle myosin (i.e., MYH9), multiple mitochondrial transcripts (e.g., MT-TL2, MT-ND1, MT-TM, MT-TD), and actin transcripts (e.g., ACTB and ACTG1); (iv) immune-related transcripts such as MHC class I molecules (e.g., B2M), interleukins (e.g., IL-6) and myosin IF (MYO1F); and (v) the lncRNAs MALAT-1 and NEAT1 (Fig 4C and Table EV2). As expected, the mRNA and lncRNA fragments that were the most consistently detected across multiple individuals were also the most highly abundant ones (Fig 4D).

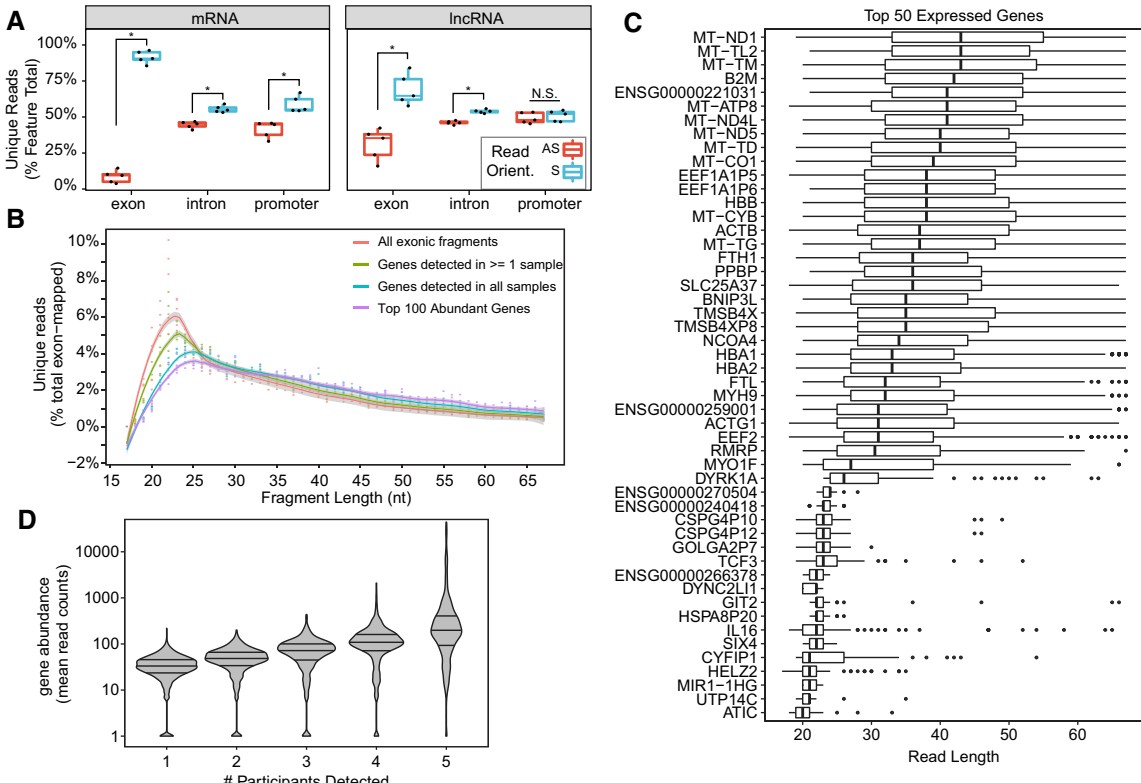

**Figure 4. Assessment of short extracellular mRNA/lncRNA fragments in human plasma using optimized library preparation and analysis methods.**

A  Boxplots showing the percentage of unambiguously annotated reads (*y*-axis) for mRNA and lncRNA exons, intron, and promoters located in the sense and antisense strand as measured by phospho-RNA-seq in plasma samples from healthy controls (*n* = 5). Boxes represent the mean ± interquartile range (IQR), and whiskers represent 1st/3rd quartile 1.5*IQR.

B  Read length distribution of exon-aligned reads in plasma samples from healthy controls (*n* = 5) prepared with phospho-RNA-seq. Read length is shown on the *x*-axis and the percent of exon-aligned reads shown on the *y*-axis. Dots represent percentages calculated for each of the five healthy control individuals. A smoothed trend line is shown and color-coded based on the categories indicated.

C  Boxplots summarize the read length (*x*-axis) distributions for the 50 most highly abundant genes (*y*-axis) across the five healthy control samples. Genes are sorted by median read length. Boxes represent the mean ± interquartile range (IQR), and whiskers represent 1st/3rd quartile 1.5*IQR.

D  Violin plots showing gene abundance expressed as mean read counts (*y*-axis) as a function of the number of participants where they were detected (*x*-axis).

## Assessment of extracellular microRNA capture from plasma by phospho-RNA-seq

After having demonstrated that phospho-RNA-seq combined with a stringent pipeline is critical for recovering mRNA and lncRNA fragments in plasma, we examined the efficiency of this strategy for capturing microRNAs from plasma by analyzing mature microRNA read counts from healthy donor plasma samples prepared with and without PNK treatment. The microRNA counts were obtained from the sRNAnalyzer alignment stage used in our exRNA processing pipeline. We found that while microRNAs were captured using the phospho-RNA-seq method, the standard small RNA-seq method yielded approximately ten times more microRNA reads (Fig EV1A). Although nearly 200 microRNAs were detected in PNK-treated samples from all five individuals, the additional coverage achieved by the standard small RNA-seq method enabled detection of 98 additional microRNAs, where detection was defined as observing them in all five individuals (Fig EV1B). We confirmed that these results were not due to differences in sequencing depth, as the standard small RNA-seq libraries were sequenced to lesser depth

(median of 9.2 million reads per individual after adapter trimming and size filtering) than the phospho-RNA-seq libraries (median of 27.9 million reads).

We found that 21 of these additional microRNAs were seen exclusively using standard small RNA-seq (i.e., they had zero reads across all five individuals with phospho-RNA-seq; Fig EV1C). Interestingly, we found some microRNAs that were exclusively captured with phospho-RNA-seq, in spite of the lower read coverage for microRNAs with this method (Fig EV1D). Closer inspection revealed that 16 of the 30 PNK-specific micro-RNAs were likely fragments of transposable or repetitive elements, or were non-functional, mis-annotated microRNAs now removed from miRBase annotations. However, excluding these artifactual microRNAs, 14 *bona fide* microRNAs were detected specifically in PNK-treated samples, indicating that a small subset of microRNAs may be present in a modified form in plasma that prevents capture with traditional methods. Overall, however, our analysis indicates that standard methods are preferable over phospho-RNA-seq when the goal is to exclusively characterize microRNA profiles in plasma.

**Pathophysiologic processes are reflected in plasma exRNA transcriptome profiles revealed by phospho-RNA-seq**

Having confirmed that the phospho-RNA-seq approach with high-stringency bioinformatic analysis enables the detection of mRNA and lncRNA fragments consistently expressed in plasma from healthy individuals under homeostatic conditions, we sought to evaluate whether dynamic (patho)physiological processes would be reflected in the expression patterns of mRNA and lncRNA extracellular fragments. To this end, we collected serial plasma samples from patients undergoing allogeneic HSCT ($n$ = 22 samples in total, from two different patients), prepared phospho-RNA-seq libraries from each time point, and performed multiplex sequencing using a HiSeq platform (Fig 5A).

We reasoned that in order for mRNA/lncRNA exRNA sequences in plasma to have potential as biomarkers, we should see patterns in specific sets of genes that correlate over time to biological processes happening within the patients. We began our analysis by using the EBSeq-HMM R package (Leng *et al*, 2015), which uses an autoregressive hidden Markov modeling strategy to test for genes that show evidence of differential expression over time (Leng *et al*, 2015). This analysis resulted in 690 (patient P04) and 275 (patient P07) genes showing significant (FDR < 0.01) evidence of dynamic changes in expression (Tables EV3 and EV4). We hypothesized that differentially expressed transcripts with similar expression patterns might have similar tissue origins or biological functions. Using an unsupervised clustering strategy from the R package, WGCNA, we identified sets of the differentially expressed genes showing concordant temporal co-expression patterns (Tables EV3 and EV4; Langfelder & Horvath, 2008). Among the differentially expressed transcripts of both patients, we found multiple transcripts known to be specific to or enriched in bone marrow (Fig 5B). The bone marrow transcripts clustered into distinct co-expression sets identified by WGCNA. In fact, a hypergeometric test identified three distinct temporal co-expression clusters for subject P04 and two for subject P07, which were significantly enriched for bone marrow transcripts. Since HSCT is a process that can be followed through the peripheral white blood cell count (WBC), we plotted abundance of the significantly enriched clusters of bone marrow exRNA transcript fragments over time along with WBC count. As shown in Fig 5C, we saw that transcript fragments corresponding to the bone marrow gene set were tracked generally with the dynamics of bone marrow reconstitution, initially declining as expected during the period of early neutropenia in the first week after transplant followed by a rise corresponding to recovery of the WBC count.

Interestingly, co-expression clusters were enriched for genes specific to or enriched in other somatic tissues. In fact, the most significantly over-represented tissue in both individuals was the liver, with nearly all liver-enriched genes grouped in the same co-expression cluster (Fig 6A and B). We wondered whether the liver-enriched transcripts corresponding to our liver gene set would track temporally with liver injury, which is common in HSCT patients, sometimes as a side effect of medications given as part of their clinical care. By plotting blood levels of serum aminotransferases (AST and ALT), two enzymes produced by liver cells that are used clinically for detecting liver injury, together with levels of exRNA liver-enriched transcript fragments over time, we saw that levels of the liver-enriched RNA fragments showed dynamic changes and

followed a similar trend as the pattern of changes in AST/ALT (Fig 6C and D). Thus, we concluded that bone marrow-specific and liver-specific exRNA transcript fragments show distinct expression patterns corresponding to known biology as measured by relevant, established clinical laboratory markers. These results provide a proof of concept that this approach can provide access to a circulating transcriptome with potential for biomarker development.

# Discussion

Interest in the study of exRNAs has been growing rapidly, both for their potential clinical application as biomarkers of disease measurable in biofluids such as blood (Schwarzenbach *et al*, 2011; Roser *et al*, 2018) and for their potential biological functions (Zhang *et al*, 2010; Hu *et al*, 2012; Shah & Calin, 2014). Most studies of exRNAs in human biofluids to date have focused on microRNAs. Unlike microRNAs, which are a small fraction of total genes, lncRNAs and mRNAs comprise a majority of the transcriptome and hold great potential as biomarkers given their exquisite tissue-specific expression (Liu *et al*, 2008; Iyer *et al*, 2015) and the fact that mRNA gene signatures in tissues have proven to be powerful biomarkers in different clinical settings (Perou *et al*, 2000; Potti *et al*, 2006; Chen *et al*, 2007; Ben-Porath *et al*, 2008). Thus, the ability to read transcriptomic information through exRNA profiles in blood plasma is important for enabling clinical applications.

However, mRNAs and lncRNAs have not been easily or consistently detectable as exRNA in blood plasma. RNAs found in plasma have generally been seen to be fragmented relative to their cellular RNA counterparts (Mitchell *et al*, 2008), and most studies have used small RNA-seq protocols that are designed to sequence microRNAs. These protocols typically employ RNA ligase-based adapter ligations, followed by reverse transcription and PCR to generate libraries for high-throughput sequencing. Such protocols commonly rely on the presence of 5′ P and 3′ OH ends on the target RNA (Hafner *et al*, 2008), which are produced by the RNase III class of ribonucleases, including the double-stranded ribonuclease Dicer that is responsible for processing precursor microRNAs to generate mature microRNAs (Knight & Bass, 2001; Ha & Kim, 2014). We hypothesized that mRNA and lncRNA transcripts in the blood circulation are likely to be acted upon by different classes of ribonucleases that may not produce ends conducive to sequencing by standard small RNA-seq library protocols. Specifically, abundant RNases in human circulation such as those belonging to the ribonuclease superfamily A (Lu *et al*, 2018) degrade RNA dinucleotide bonds leaving a 5′ OH and 3′ P product (Cuchillo *et al*, 2011), thus rendering cleavage products unsuitable for standard ligation-based library preparation protocols.

We sought to test this hypothesis using both ground truth samples of synthetic RNA pools with variable 5′ and 3′ end modification states and biological (plasma) samples. Our results clearly showed that PNK treatment vastly increased the recovery of fragments either lacking a 5′ P or having a 3′ P. This is consistent with the properties of PNK, which has both 5′ kinase and 3′ phosphatase activities (Richardson, 1965; Novogrodsky & Hurwitz, 1966; Cameron & Uhlenbeck, 1977). Moreover, these results highlight that standard ligation-based small RNA-seq approaches are not well suited to characterize exRNA beyond microRNAs and other

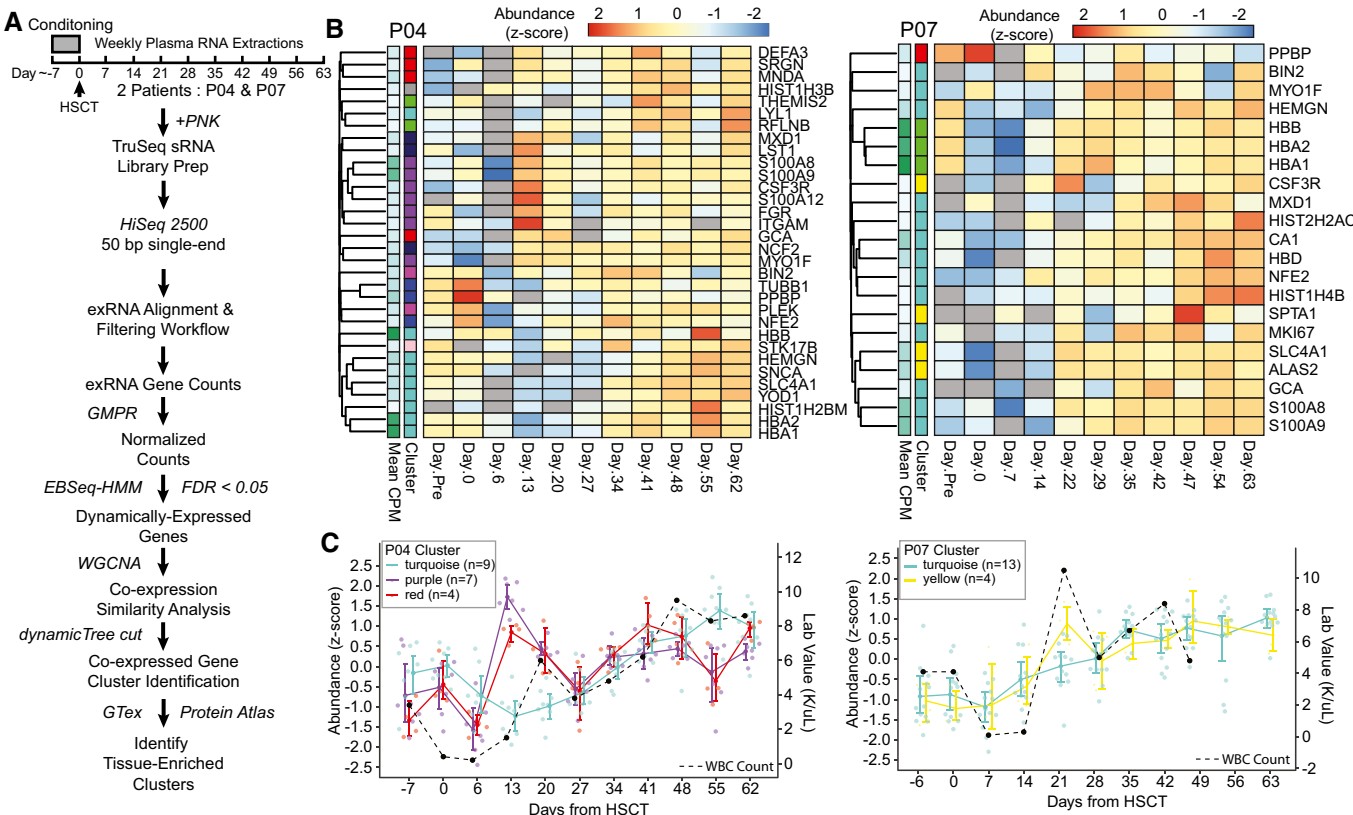

**Figure 5. Relationship between temporal patterns of dynamically co-expressed plasma mRNA/lncRNA fragments and bone marrow recovery in patients undergoing allogeneic HSCT.**

A   Schema depicts timing of the serial sample collection, experimental methodology, and bioinformatics analyses performed to assess gene co-expression and tissue enrichment.

B   Heatmaps show the expression patterns of bone marrow-enriched genes that were detected in patients P04 (left) and P07 (right) data sets, and found differentially abundant by EBSeq-HMM (FDR < 0.01). Geometric mean of pairwise ratio (GMPR)-normalized read counts were centered gene-wise to have a mean of 0 and standard deviation of 1. The "cluster" row annotations indicate the co-expression cluster identified by WGCNA. P04 clusters, turquoise, purple, and red, and P07 clusters, turquoise and yellow, were significantly enriched for bone marrow transcripts (hypergeometric test; FDR < 0.01).

C   Graphs show the expression patterns for the co-expression clusters significantly enriched for bone marrow-enriched transcripts. Individual points are shown for each gene in the P04 (left) and P07 (right) co-expression clusters, and are colored according to the cluster IDs. Colored lines indicate the mean expression of each cluster, and error bars represent a bootstrapped (B = 1,000) 98% CI of the mean. Black dashed lines indicate the white blood cell counts obtained from laboratory results measured on the same day.

---

sequences sharing the same end chemistries. Our method revealed that human plasma contains abundant mRNA and lncRNA transcript fragments, corresponding to thousands of human genes. Thus, mRNA and lncRNA fragments are a substantial component of the extracellular transcriptome in human plasma. Our incorporation of synthetic RNA pools as a ground truth reference was especially important in our study, as it allowed us to demonstrate clearly that lack of a 5′ P and presence of a 3′ P are both impediments to recovery of these fragments by standard small RNA-seq.

Although PNK treatment enabled much greater recovery of sequences lacking 5′P and/or 3′OH, PNK is known to exhibit target nucleic acid sequence-dependent biases in its kinase and phosphatase activities (Lee *et al*, 2013). These very likely influence the distribution of sequences we are able to recover with the current iteration of the phospho-RNA-seq method, limiting the accuracy of estimating the relative abundance of different mRNA/lncRNA fragments within a sample. Incorporation of an OptiKinase step prior to

PNK has been reported to mitigate bias observed when using PNK alone (Lee *et al*, 2013), and could be tested in future iterations of the phospho-RNA-seq method. In addition, future studies using large, diverse sequences of RNA fragments with varying 5′ and 3′ phosphorylation states could be used to deeply characterize sequence-dependent biases of PNK treatment in the context of phospho-RNA-seq for recovery of mRNA/lncRNA fragments. Finally, the use of downstream library protocols that have less intrinsic bias (Giraldez *et al*, 2018) than the one used here (TruSeq) has potential to increase the overall recovery of unique mRNA/lncRNA fragments, as well as to improve the accuracy of estimating relative abundance of different fragments within a sample.

A key lesson learned from our study is the importance of a highly stringent data analysis for accurate identification of mRNA and lncRNA fragments from phospho-RNA-seq sequence data. This is because these relatively short sequences frequently align to multiple locations in the genome. Thus, short fragments of RNAs arising

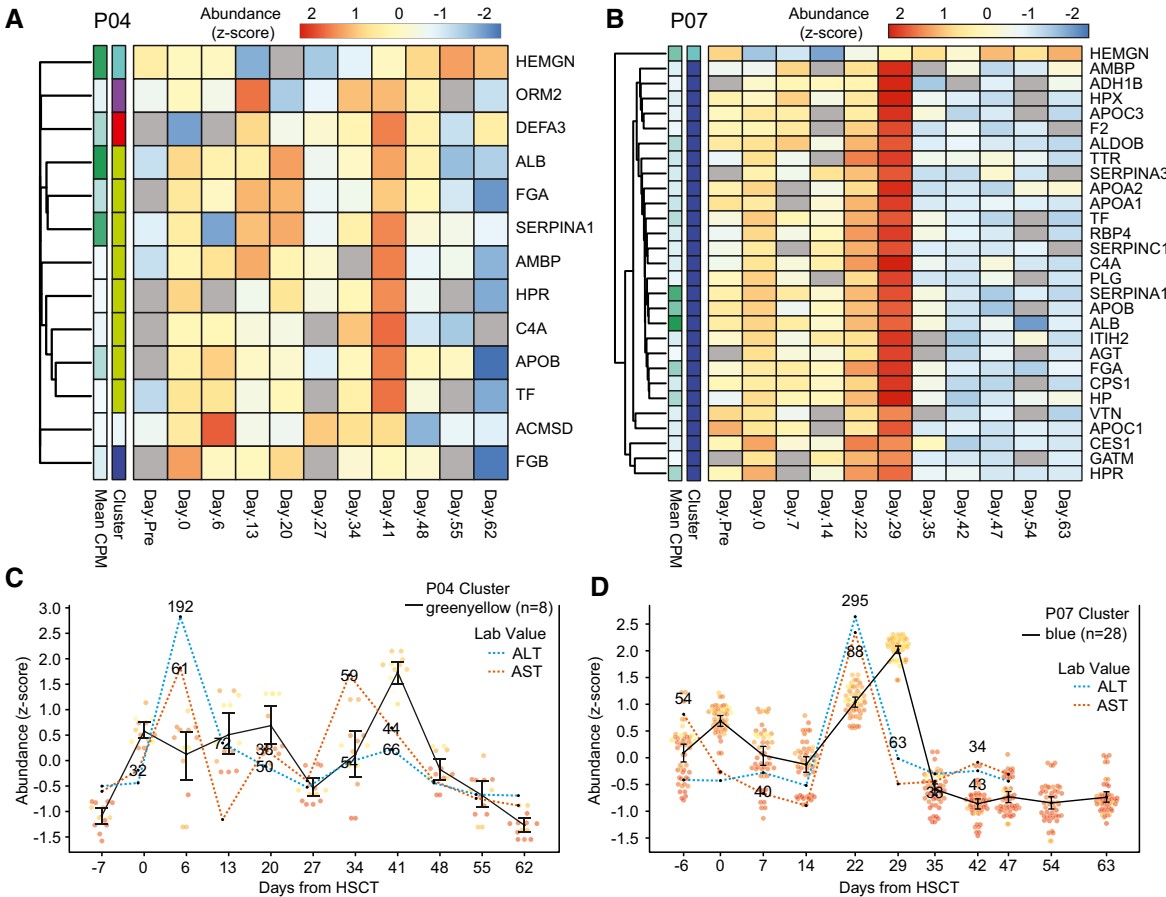

**Figure 6. Relationship between temporal patterns of dynamically co-expressed plasma mRNA/lncRNA fragments and liver injury in patients undergoing allogeneic HSCT.**

A, B Heatmap of liver-enriched genes detected and found differentially abundant by EBSeq-HMM (FDR < 0.01) in (A) P04 and (B) P07 samples. The "cluster" row annotations indicate the co-expression clusters identified by WGCNA. P04 cluster, greenyellow, and P07 cluster, blue, were significantly enriched for liver-specific and enriched transcripts (hypergeometric test; FDR < 0.01).

C Graph shows the expression for genes in the P04 co-expression cluster, greenyellow, along with AST and ALT laboratory values taken on the same day.

D Graph shows the expression for genes in the P07 co-expression cluster, blue, along with AST and ALT laboratory values.

Data information: (C, D) Points represent the z-transformed $\log_2$ read counts for each of the genes in the cluster (P04 greenyellow: $n = 8$; P07 blue: $n = 28$). The black lines indicate the mean expression of each cluster, and error bars represent a bootstrapped (B = 1,000) 98% CI of the means. Red and blue dashed lines represent laboratory values for AST and ALT liver enzymes, respectively. AST and ALT levels were centered to a mean of 0 and standard deviation of 1 for plotting. The actual laboratory values are shown for all elevated readings (AST > 30; ALT > 35).

from repetitive DNA transcription or ribosomal RNA fragments, for example, can spuriously align to mRNA and lncRNA exons. To avoid these false-positive calls, we developed a three-step filtering pipeline specifically designed for the reliable identification of mRNA and lncRNA fragments. To date, pipelines for the analysis of small RNA-seq data such as the exceRpt pipeline (http://genboree.org/the Commons/projects/exrna-tools-may2014/wiki/Small_RNA-seq_Pipe line) have been designed with a predominant focus on microRNA annotation. Perhaps not surprisingly, we found that when applied to phospho-RNA-seq data analysis for mRNA/lncRNA identification, there was a high rate of false-positive annotation. It is possible that this phenomenon may have affected prior results of plasma exRNA profiling using non-PNK-based approaches which, although focused primarily on microRNAs, also commented on the finding of mRNA and lncRNA transcripts in plasma (Huang *et al*, 2013; Yuan *et al*,

2016; Max *et al*, 2018). Future work could determine the extent to which reports of mRNA and lncRNA fragments from libraries not incorporating PNK may reflect false-positive mis-annotation, due to limitations of bioinformatic analysis pipelines. We propose that the high-stringency pipeline we describe here is one approach for mitigating the false-positive rate, and there may be further bioinformatic approaches that can also be developed.

We must acknowledge, however, that the strict filtering used to remove repeats, although effective in reducing false positives, also potentially removes some valuable information and can lead to false negatives. Future work could utilize additional features such as high-confidence alignments throughout the exons of a gene in the proper orientation relative to the transcript. This may provide enough evidence to enable us to capture reads aligning to embedded repetitive sequences in *bona fide* mRNA/lncRNA transcripts, for example.

PNK is a well-characterized enzyme frequently used to create appropriate and homogeneous RNA ends before long RNA sequencing. Specifically, this enzyme has been used in long RNA sequencing protocols to render tissue RNA suitable for adapter ligation after experimental heat- or alkali-based fragmentation (Lamm *et al*, 2011; Lee *et al*, 2013). PNK has also been included in protocols aimed to identify 5′PPP moieties that correspond to transcriptional start sites in bacteria (i.e., differential RNA-seq) (Vvedenskaya *et al*, 2015). In contrast, PNK is not generally used as part of small RNA protocols and it has not been specifically evaluated as a strategy for revealing novel mRNA/lncRNA fragment sequences in biological samples such as plasma, which are missed by standard ligation-based small RNA protocols.

We applied phospho-RNA-seq to analyze longitudinal plasma samples from HSCT patients for two reasons. The first was to provide additional validation for our methods and data analytic pipeline, in that finding gene signatures in plasma that correspond to known time-dependent biological changes in patients would make it highly unlikely that the mRNA/lncRNA fragments we found were spurious and due to mis-annotation, for example. The second was to establish the proof of concept that mRNA/lncRNA fragment gene signatures in plasma correlate with human biology, indicating the potential of this method as a new, broadly applicable liquid biopsy approach. As shown here, we found through an analysis beginning with temporal plasma exRNA gene expression profiles and progressing to identify tissue-specific gene sets over-represented in the dynamic profiles, that specific gene signatures corresponding to bone marrow and liver both were tracked in a manner that paralleled the dynamic processes of bone marrow reconstitution and liver toxicity, respectively.

The results in HSCT patients indicate that the phospho-RNA-seq approach, when combined with high-stringency bioinformatic data analysis, could be applied to developing biomarkers for a range of diseases in which tissue injury figures prominently. For example, this could include evaluation of the cause of hepatic injury, where plasma liver transcript signatures might provide more detailed information about underlying etiology than standard liver enzyme measurements in blood, thus assisting with differential diagnosis. Similarly, plasma transcriptome signatures may inform about damage to other organs, as can occur as a result of myocardial ischemia, autoimmune diseases, placental dysfunctions during pregnancy, and cancer treatment (e.g., chemotherapy, immunotherapies). We speculate that cancer tissue-specific transcript gene sets could also be developed for detection and longitudinal monitoring of a variety of cancer types.

Although most prior plasma exRNA studies have focused on microRNAs, mRNAs in plasma have been reported using methodology other than phospho-RNA-seq. In particular, a study using a long RNA-seq random primer cDNA synthesis protocol detected mRNAs and lncRNAs in plasma (Koh *et al*, 2014). Given that the minimal length of RNAs generally captured by that protocol is larger than what we observed with phospho-RNA-seq, we expect that approach is unlikely to have captured many of the mRNA/lncRNA fragments found in our study. Taken together, the results suggest that exRNA transcripts of a broad range of lengths might coexist in the human circulation. It remains an open question at this point as to what fraction of the plasma exRNA transcriptome is present as shorter fragments of the form revealed in our study, or as presumably longer RNAs reported in prior work (Koh *et al*, 2014). The shape of the

exRNA size distribution observed in our data would indicate that longer fragments (e.g., > 100 bases) may exist, but may be a minority. However, this question would need to be addressed in future studies to obtain a more accurate characterization of the exRNA transcriptome present in blood and other biofluids.

Although the phospho-RNA-seq strategy allowed us to expand the spectrum of sequences that can be detected in plasma (i.e., sequences lacking a 5′ P and/or presenting a 3′ P end), there are certainly still limitations of this approach. Although we expect that most of the cleavage products derived from mRNA/lncRNA in circulation will present 5′ OH and 3′ P due to the abundance of superfamily A RNases in plasma, it is possible that there are exRNAs in plasma with end groups that are not addressed by our methodology (e.g., 5′ cap). Second, it is worth mentioning that the efficiency of phospho-RNA-seq for identifying mRNA/lncRNA fragments is affected by the recovery of other abundant plasma fragments such as ribosomal RNA and Y RNA that can dominate the sequencing data, thus reducing the depth of sequencing available for detecting mRNA and lncRNA fragments. To this end, we foresee that in the future, the sensitivity for detecting relevant mRNA and lncRNA fragments in plasma might be improved by designing specific methods for depleting abundant, undesired fragments such as fragments of rRNAs and Y RNAs, or for enriching fragments corresponding to panels of selected transcripts. It is clear that there is still room for method improvement and there may be many more mRNA and lncRNA fragments in plasma than we have identified here. It is also worth noting that new ligation-free strategies have been recently developed for small RNA-seq (Turchinovich *et al*, 2014). Future studies will be required to determine the efficiency and effectiveness of these approaches for detecting mRNA and lncRNA fragments in circulation. Moreover, taking into account our results, we envision that sequence data analytic pipelines specifically designed for reliable analysis of mRNA/lncRNA fragments, such as the one described here, will be required for the analysis of exRNA sequencing data generated with these strategies.

In addition to their potential utility as biomarkers in human disease, there also remains the question of whether these circulating extracellular mRNA and lncRNA fragments play any physiological role. Multiple intriguing but controversial studies have suggested that some extracellular microRNAs serve a functional role by mediating cell–cell communication (reviewed in Tkach & Thery 2016). Another recent report showed that the non-coding RNA RN7SL1 can be transferred between cells via an extracellular form, resulting in activation of an innate immune response in the recipient cell (Nabet *et al*, 2017). Although it is unlikely that the mRNA fragments we observed retain sufficient information to direct meaningful protein translation, we speculate that there may be some specific fragments that could be functional. They might serve as an intercellular signal through their own RNA sequence, or perhaps through interactions with associated RNA binding proteins that might be bound to the fragments.

In summary, our results highlight that there is greater complexity of the extracellular transcriptome in human biofluids than previously known and that phospho-RNA-seq can provide access to transcriptomic signatures in plasma that are inaccessible by standard small RNA-seq methods. The methodology presented here provides access to a new class of extracellular RNAs for development as liquid biopsy biomarkers for a variety of diseases. In addition, it may be useful to investigate this technique for application to other

settings where RNA is highly degraded, including formalin-fixed paraffin-embedded archival tissue specimens, as well as extremely old specimens of cells or tissues, where RNA may likewise be present in highly fragmented form.

# Materials and Methods

## Synthetic reference sample

A synthetic equimolar pool containing 476 synthetic RNA oligonucleotides was prepared in an RNase-free environment and working on ice to minimize degradation. The pool was prepared by combining (i) 286 human microRNAs and (ii) a set of 190 additional, custom-synthesized RNA oligonucleotides, to generate the pool in which each of the 476 RNA oligonucleotides is present at equimolar concentration. The latter set of 190 RNA oligonucleotides comprises micro-RNAs and non-microRNA sequences of varied length from 15 to 90 nt, which were synthesized, HPLC-purified, and quantified spectrophotometrically by IDT (Coralville, IA, USA). The pool of RNA oligonucleotides is available to qualified investigators seeking to reproduce the synthetic equimolar for non-commercial purposes, by request of the corresponding author (as long as supplies last). The resulting equimolar pool was aliquoted in prelabeled DNA-, DNase-, RNase-, and pyrogen-free screw-cap tubes with low adhesion surface and stored immediately at −80°C. The complete list of RNA sequences comprising the equimolar pool is provided in Table EV1.

## Biological samples

Plasma samples from five healthy donors and serial plasma sample from two patients undergoing allogeneic HSCT were collected in 10-ml K2EDTA plasma tubes (Vacutainer 366643; Becton Dickinson, Franklin Lakes, NY, USA) and processed within one hour of blood draw following a two-centrifugation protocol to obtain platelet-poor plasma as previously described (Cheng *et al*, 2013): (i) 3,400 *g* at room temperature for 10 min with high brake; and (ii) 1,940 *g* at room temperature for 10 min without brake. Plasma was stored at −80°C until RNA isolation. The University of Michigan IRB approved the study protocol to consent participants and collect samples. Informed consent was obtained from all subjects, and the samples were subsequently de-identified before distributing to the laboratory personnel generating the libraries. The studies conformed to the principles set out in the WMA Declaration of Helsinki and the Department of Health and Human Services Belmont Report.

RNA was isolated from 200 μl of plasma using the miRNeasy Mini Kit (Qiagen, Hilden, Germany) according to the manufacturer's protocol with the following modifications. Plasma samples were mixed with five sample volumes of QIAzol reagent and vortexed for 10 s. Samples in QIAzol were incubated at room temperature for 5 min to inactivate RNases. Next, 0.2 volumes of chloroform were added to each sample. At that point, the manufacturer's protocol was followed.

## Library preparation and sequencing

The input for library preparation was 10 femtomoles of RNA for the synthetic equimolar pool and 5 μl of eluted RNA for the biological plasma samples.

Standard ligation-based small RNA libraries were prepared using the TruSeq small RNA kit (Illumina, San Diego, CA, USA) according to the manufacturer's instructions. Size selection was performed using pre-cast 6% acrylamide gels (Invitrogen, Carlsbad, CA, USA) including all products from 140 to 200 bp plus any additional visible bands of greater size. To perform phospho-RNA-seq, synthetic and plasma RNA samples were pretreated with T4 polynucleotide kinase (NEB, Ipswich, MA, USA) using an RNA input of 7 μl in a final reaction volume of 10 μl and incubated at 37°C for 30 min following the manufacturer's instructions. After the enzymatic treatment, synthetic RNA samples were heat-inactivated at 65°C for 20 min and biological RNA samples were purified by performing sequential washes in silica columns (Zymo, Irvine, CA, USA): (i) 900 μl of buffer RWT (Qiagen, Hilden, Germany); (ii) 900 μl of buffer RPE (Qiagen, Hilden, Germany); (iii) 900 μl of ethanol 200 proof, molecular biology grade (Fisher Scientific, Waltham, MA, USA); and (iv) 900 μl of 80% ethanol. Libraries were then prepared using the TruSeq small RNA kit according to the manufacturer's instructions. Size selection was performed as described above. For the libraries generated from patients undergoing HSCT, we narrowed the range of size selection to 140–165 bp to reduce the abundance of contaminants such as Y RNAs.

Libraries were multiplexed and sequenced using the Illumina NextSeq 500 (synthetic equimolar pool) and Illumina HiSeq 2500 (healthy controls and patients undergoing allogeneic HSCT) specifying 75 and 50 bp single-end runs, respectively.

## Computational methods

### exRNA processing pipeline

TruSeq adapters and stop oligo sequences were trimmed with cutadapt (v 1.91) using processing steps adapted from the sRNAnalyzer workflow (Martin, 2011; Wu *et al*, 2017). The sRNAnalyzer framework was also adapted to align adapter-trimmed reads 15 nt and longer to several sequence databases containing known small RNA families and contaminant sequences (Wu *et al*, 2017). A table with descriptions of the included sequence databases is provided in Appendix Table S2. Up to two mismatches were allowed in the alignment.

Reads that had no valid alignments to the various endogenous non-coding RNA sequences and contaminant databases were aligned to the human genome (GRCh38) using STAR (v 2.5.0A; Dobin *et al*, 2013). The following parameters were altered from default:

outFilterMultimapNmax = 1,000,000; outFilterMismatchNoverLmax = 0.1; outFilterMatchNmin = 15; outFilterMatchNminOverLread = 0.9; outMultimapperOrder = Random; outSAMtype = BAM Unsorted; outReadsUnmapped = Fastx; outSAMattributes = All; outSAMprimaryFlag = AllBestScore; alignIntronMax = 1; alignIntronMin = 2; alignSJDBoverhangMin = 999

These parameters remove the splicing-aware alignment capability and limit the extent of "soft-clipping" at the ends of the alignment.

### Synthetic pool library analysis

Illumina NextSeq reads from equimolar synthetic pool libraries were processed to trim adapters, remove low-quality bases, and filter

short reads using the exRNA processing pipeline described above. Reads as short as 15 nt were allowed for detection of the shortest oligos in the pool. STAR (v2.5.0A) was used to align the preprocessed reads to the equimolar pool sequences. The following alignment parameters were altered from default: outFilterMultimapNmax = 1000000; outFilterMismatchNoverLmax = 0.1; outFilterMatchNmin = 15; outFilterMatchNminOverLread = 0.9; outMultimapperOrder = Random; outSAMtype = BAM Unsorted; outSAMunmapped = Within; outSAMattributes = All; outSAMprimaryFlag = AllBestScore; alignIntronMax = 1; alignIntronMin = 2; alignSJDBoverhangMin = 999. Read alignments were loaded into R for processing and analysis. Alignments were further filtered, requiring the alignment to match at least 90% of the synthetic pool sequence in the sense orientation, and be at least 15 nt in length after soft-clipping. Read counts were scaled for multi-mapping, dividing the counts by the number of valid alignments obtained from the "NH" tag in the bam alignment file.

### MicroRNA analysis

Read counts for mature human microRNAs were taken from the sRNAnalyzer ".profile" counts generated in the processing pipeline. Only read counts with ≤ 1 mismatch in the sense orientation were used. Read counts for each microRNA were summed across technical replicates. Library size-adjusted read counts were calculated as counts per million.

MicroRNA genomic coordinates (GRCh38) were obtained from miRBase V22 (http://www.mirbase.org/). Accession numbers missing from the coordinate gff files corresponded to microRNAs removed from miRBase due to lack of functional evidence and were annotated as "missing from miRBase". Transposon and repeat-associated microRNAs were annotated by overlapping microRNA coordinates with the RepeatMasker coordinates (UCSC Genome Browser; hg38), requiring a minimum of 1 nt overlap in either orientation.

### Multi-mapping scaling and gene quantification

Read counts were weighted using a strategy similar to that employed by CSEM, which gathers mapping information from neighboring read alignments to weight read counts toward loci with the most unambiguous mapping information (Chung *et al*, 2011). Our strategy differs in that we (i) gather mapping information from neighboring read alignments across all samples in the cohort, (ii) restrict our search to directly overlapping fragments, and (iii) retain the mapping ambiguity information to allow identification of commonly co-mapping genes. To do this, a bipartite network was created using the R package, igraph, to connect reads with all overlapping clusters of mapped loci (Csardi G, Nepusz T: The igraph software package for complex network research, InterJournal, Complex Systems 1695. 2006). All connected components were identified, using the mapping ambiguity information from all connected reads to weight reads more strongly to those regions with more unambiguously mapped reads. Read alignments were annotated for overlap with (i) GENCODE transcripts, (ii) GENCODE various endogenous non-coding RNAs, and (iii) RepeatMasker annotation coordinates (UCSC genome browser). All alignments were removed for any read aligning to GENCODE various endogenous non-coding RNA or RepeatMasker loci (minimum 1 bp overlap in either orientation).

### Comparison with exceRpt pipeline

The exceRpt small exRNA analysis pipeline (v 4.6.2) implemented on the Genboree Workbench (http://genboree.org/java-bin/login.jsp) was used to process and analyze healthy control samples for a healthy control (ULMC135), prepared using both the standard TruSeq small RNA library protocol and the modified phospho-RNA-seq method. Default exceRpt pipeline parameters were used, except to set the minimum read length to 16, and to specify TruSeq small RNA adapters for trimming. Read counts from GENCODE transcripts were obtained from the post-processed exceRpt output files and were compared with the ULMC135 read counts from our pipeline collected at: (Stage 1) after adapter trimming and size filtering, (Stage 2) after sRNAnalyzer alignment and contaminant removal, and (Stage 3) after genome alignment and removal of RepeatMasker and various endogenous non-coding RNA coordinate-based annotations. Reads from each stage were aligned to the human genome with STAR, using the same alignment parameters described above. These parameters were largely copied from those used by the exceRpt pipeline to make the alignments as comparable as possible. Comparison was limited to genes detected by both pipelines. Because the exceRpt pipeline output included only summarized gene abundance, the percent of various endogenous non-coding RNA or repeat-aligned reads were based on the alignments from our pipeline. An alignment was considered "sRNA or Repeat-Aligned" if any alignments for that read overlapped RepeatMasker or GENCODE various endogenous non-coding RNA coordinates (minimum 1 nt overlap on either strand). Fragments were summarized at the gene level using multi-mapping-weighted exon-aligned fragments, comparable to that used by the exceRpt pipeline for gene-level quantification.

### Cell-free RNA enrichment in coding and non-coding regions of mRNA and lncRNA transcripts

Exon, intron, and promoter (2 kb upstream + 0.2 kb downstream nt) coordinates were extracted from GENCODE (v27) protein-coding and long non-coding RNA annotations. RepeatMasker and sRNA-filtered genomic alignments from the five healthy individuals were intersected with these coordinates, requiring a minimum of 1 bp of overlap in either orientation. Ambiguous annotations were allowed, but counted only once per unique combination of read and feature. Read alignments were considered "Sense" or "Antisense" based on the relative orientation of the read alignment and the mRNA or lncRNA feature. Read length distributions and gene abundance were calculated based on sense-aligned exonic reads.

### Analysis of hematopoietic stem cell transplantation cohort

HiSeq reads from P04 and P07 HSCT patients were processed and filtered as described above. Gene-level counts were calculated separately for P07 and P04 samples, using the multi-mapping-weighted read counts from mRNA and lncRNA exon-aligned read fragments. The resulting gene count matrices were normalized across samples using a robust geometric mean of pairwise ratio (GMPR) method, suitable for sparse data sets (Chen *et al*, 2018). The GMPR-calculated size factors were provided as input to the R package, EBSeq-HMM, which employs an autoregressive hidden Markov modeling strategy to identify genes with non-static expression dynamics over the course of the time series (Leng *et al*, 2015). EBSeq-HMM was run separately for P04 and P07 samples. An initial run was performed

using a low number of iterations ($n = 5$) to test a range of fold-change estimates (1.0–2.0, by 0.2). The estimate that maximizes the log likelihood was then used for a second run of the algorithm with a higher number of iterations (100). Significantly altered genes were selected at an FDR cutoff of 0.01 (Leng *et al*, 2015). GMPR-normalized read counts from the significantly altered genes were clustered using the WGCNA workflow (Langfelder & Horvath, 2008).

*Tissue enrichment*

Databases of tissue-enriched genes were obtained from GTex and Human Protein Atlas data curated by the TissueEnrich R package (Jain & Tuteja, 2018). Significant enrichment was determined with a hypergeometric test and using a background of all genes used as input to EBSeq-HMM analysis (Leng *et al*, 2015).

# Data and code availability

Sequencing data reported here are available at GEO, under the superseries, GSE126051 (https://www.ncbi.nlm.nih.gov/geo/query/acc.cgi?acc=GSE126051). R code and processed data files are available on GitHub (https://github.com/rspengle/phosphosRNAseq_Manuscript_Analysis).

**Expanded View** for this article is available online.

## Acknowledgements

We thank T. Churay for assistance in obtaining clinical specimens; X. Cao and A. Chinnaiyan for assistance with sequencing of healthy control libraries; E. Sandford for assistance in specimen management; J. Vandesompele, K.E.A. Max, and T. Tuschl for helpful discussions; J.S. Rozowsky, R. Kitchen, S.L. Subramanian, W. Thistlethwaite, and A. Milosavljevic for facilitating access to the exceRpt pipeline; and K. Wang for providing synthetic microRNA oligonucleotides. We acknowledge funding support from the NIH Extracellular RNA Communication Common Fund grants: U01 grants HL126499 to M. Tewari and HL126496 to D.J.G., and from the A. Alfred Taubman Medical Research Institute (Grand Challenge Award) to M. Tewari and S.W.C. Research reported in this publication was also supported by the National Cancer Institute of the NIH under Award Number P30CA046592 by the use of the following Cancer Center Shared Resource at the University of Michigan: DNA Sequencing. M.D.G. acknowledges support from a Precision Health Scholar Award from the University of Michigan Precision Health Center and a Juan Rodes contract (JR18/00026) funded by the Spanish Institute of Health Carlos III from the Ministry of Economy and Competitiveness (co-funded by European Social Fund (ESF)). D.J.G. also acknowledges a special technology support award and partial funding from the Pacific Northwest Research Institute to his laboratory. The content is solely the responsibility of the authors and does not necessarily represent the official views of the National Institutes of Health.

## Author contributions

MDG and MTe conceived of the project. MDG and AE designed and performed the experimental work. MDG, RMS, AE, SWC, DJG, and MTe interpreted the results. MDG, RMS, and MTe wrote the manuscript. RMS designed and implemented the computational pipeline and performed the data analysis. MTe, SWC, and MTu designed the human subject studies. MTu and AJG carried out human subject studies, and AJG performed clinical data abstraction. All authors read the manuscript and provided input.

## Conflict of interest

The authors declare that they have no conflict of interest.

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
