## [Review Process File · The EMBO Journal]

Phospho-RNA-seq: a modified small RNA-seq method that reveals circulating mRNA and lncRNA fragments as potential biomarkers in human plasma

Maria D. Giraldez, Ryan M. Spengler, Alton Etheridge, Annika J. Goicochea, Missy Tuck, Sung Won Choi, David J. Galas, Muneesh Tewari

Review timeline:

Submission date:	5th Feb 2019
Editorial Decision:	21st Mar 2019
Revision received:	7th Apr 2019
Editorial Decision:	11th Apr 2019
Revision received:	14th Apr 2019
Accepted:	15th Apr 2019

Editor: Karin Dumstrei

Transaction Report:

1st Editorial Decision

21st Mar 2019

Thanks for submitting your manuscript to The EMBO Journal. I am sorry about the delay in getting back to you, but I have now received the three reports on your paper.

As you can see from the comments while referees #2 and 3 are supportive of the study, referee #1 is not convinced that we gain enough new insight. I have looked carefully at all the points and while I see the points that referee #1 are raising, I also find the method development aspect of the study important and I also really appreciate the careful side by side comparison and validation of the method. I would therefore link to invite a revised version.

Regarding the specific concerns raised by referee #1:

Points 1 and 2: I would suspect you would have insight into this from the reference library that you use in Figure 1

Point 3: Seems like a relevant issue, but let's discuss this point

Point 4: please comment

Point 5: please comment on these points - we don't need the lncRNA/mRNAs to be validated in a second cohort. For us the importance is the method development - just make sure you provide a carefully discussion about this point.

The remaining points should be straightforward to address. You can use the link below to upload the revised manuscript.

REFeree REPORTS:

Referee #1:

General summary and opinion about the principle significance of the study, its questions and findings

The main significance of the study by Giraldez et al. "Phospho-sRNA-seq reveals extracellular mRNA/lncRNA fragments as potential biomarkers in human plasma" is the development of a novel method for profiling extracellular messenger RNAs (mRNA) and long noncoding RNAs (lncRNA) in plasma, for a potential use as biomarkers.

The authors added an additional step prior to library prep method, which modifies the 5' and 3' ends of small RNAs which do not have these residues naturally so that these ends will contain 5'-phosphate and 3'-OH after modification that will allow adapter ligation.

As a first proof-of-concept, the authors show that PNK treatment increases sequencing depth in sequences that lack 5'-phosphate, and sequences that lack 5'-phosphate and in addition have 3'-phosphate, from an equimolar pool. In contrast, it did not affect sequences with 5'-phosphate, most of which are miRNAs (Fig. 1). They further demonstrated that when combined with a custom, high-stringency bioinformatic pipeline, the new method, termed Phospho-sRNA-seq, reduces false positive mRNA/lncRNA fragments in plasma (Fig. 2). They further strengthened this finding by re-evaluating the top transcripts called by a standard small RNA-seq analysis pipeline with their custom pipeline (Fig. 3). In Figure 4 they demonstrate the enrichment of sense vs anti-sense transcripts in aligned reads, for both mRNAs and lncRNAs and the length distribution of the reads. It also shows the annotations for the 50 most abundant genes in the plasma of healthy patients. Figures 5 and 6 show longitudinal changes in mRNA/lncRNA fragments in the plasma of two patients undergoing bone marrow transplantation (BMT). Focus is given to bone-marrow enriched and liver-enriched transcripts and comparing the time course of changes in gene clusters with that of changes in known laboratory markers for response to the BMT (white blood cell count) and liver injury (ALT and AST plasma levels). Authors then claim that temporal changes in gene clusters correspond to changes in the well-established laboratory markers.

The study was technically well-conducted and the data seem to be solid. There is probably room / market for the new variant of RNA NGS library prep. However it is a relatively small modification and hence the authors go so demonstrate the biological value of the new method in Figure 5,6. However these clinically-related data are too preliminary. The analyses performed are not sufficient to support the value of the method and the clinically-related parts. Additional experiments should have been performed to this end, as detailed below.

Major concerns

1. Experimentally assess potential biases created by the PNK treatment. Does PNK treatment have a preference for phosphorylating/dephosphorylating of certain sequences? I am specifically concerned about the fact that PNK may act on one site in those sequences with 3'-OH (i.e. only on the 5'-OH, as there is no phosphate in the 3') and on two sites in the sequences with a 5'-OH and 3'-P, which may add to the bias.
2. Might PNK treatment affect adapter ligation properties? Would it not make more sense to use a library prep method that mitigates such potential bias (and not TruSeq), as the authors themselves demonstrated in their previous work (Comprehensive multi-center assessment of small RNA-seq methods for quantitative miRNA profiling, *Nat. Biotech.*, 2018)?
3. In order to establish the superiority over commercially available methods, additional small All of the QC measurements (i.e. % of aligned reads mapped to mRNA/lncRNA, CPM, number of reads above a certain threshold etc.) should be assessed in more than one library prep platform RNA seq platform (not only Truseq.).
4. In Figure 4C, it seems that about half of the 50 most highly abundant genes greatly vary in length (looking at the dots to the right of the boxplots). What is the reason for that?

5. The claim that specific exRNA transcript fragments might provide access to a circulating transcriptome with potential for biomarker development is true in general. However the data and analysis provide is not mature enough to support the specific claim. Thus, for data presented in Fig. 5C and 6C, D:

- a. The figures only show a general trend, but there is no report on correlation between levels of mRNAs/lncRNAs to levels of ALT/AST and numbers of white blood cells.
- b. The levels of the genes in P04 and P07 cluster do not seem to change longitudinally to the same extent as ALT/AST and white blood cell, and have more moderate peaks. Therefore, it does not seem suitable for tracking time-dependent changes.
- c. The predictive power of lncRNAs and mRNAs should be determined in a larger and independent set of patient samples and healthy controls (replication study) and compared to predictive power of well-known and well-established biomarker, in order to support the main claims of the authors.

Minor concerns that should be addressed

1. The author state that miRNAs are "exceptionally stable in plasma" (p.2), and further state that the stability of mRNAs and lncRNAs in plasma is questionable ("mRNAs and lncRNAs, if truly stable in blood plasma at all...", p. 3). If so, then explain why looking at mRNAs and lncRNA as markers is of value over measuring miRNAs?
2. Explain why the first experiments in equimolar synthetic pool was performed with the NextSeq platform and subsequent experiments in plasma were performed in HiSeq.
3. Why was the input material for library prep determined by volume (5µl) and not RNA amount in ng?
4. In figure 1B, show PNK- box plot at the left hand side, so that it is clear that PNK treatment increases the CPM for sequences with 5'-OH and 3' P/OH.
5. In Figure 2E, only absolute numbers of uniquely mapped and multi-mapped reads are presented. It seems that both are increased by PNK. Thus, is there a change in the % of uniquely and multi-mapped reads as well?
6. In p. 9, the authors refer to "Figure 4D and Appendix Table S3" but it might be referring to Figure 4C that shows the top 50 expressed genes.

Summary: The manuscript falls short of the required novelty and its overall contribution to the field is premature and hence below what I wish to see at the EMBO Journal.

Referee #2:

This paper reports an improved method for cDNA cloning, sequencing, and analysis of RNA transcript sequences in circulation in human blood plasma. Methods for sequence analysis of circulating microRNAs have been well established, but previously no good methods were available for sequencing mRNA or lncRNA transcripts in circulation, apparently for the reason that, unlike microRNAs, such degradation products of longer transcripts would be expected to lack 5' phosphate, and often contain a 3' phosphate. The authors devised a method that employs T4 polynucleotide kinase treatment of RNA extracted from plasma, followed by standard ligation-based cDNA cloning, and they show that their method substantially enhances the yield of sequence reads from the sense strands of mRNAs and lncRNAs. They employ their method in the context of longitudinal studies of patients undergoing bone marrow transplants, with the results showing apparent dynamic signatures of bone marrow and liver transcripts. The authors also demonstrate the importance of employing stringent computational filters to mitigate the false-positive effects cause by highly abundant and repetitive sequence reads This is a valuable new method, well documented by convincing data. Publication in EMBO Journal is encouraged. There are only two minor critiques:

- 1) Figure 6C and 6D do not seem to be specifically cited in the text.
- 2) Please provide complete reaction conditions for the T4 PNK treatment.
- 3) It seems like the authors may have missed an opportunity to address whether there may be a population of unphosphorylated microRNA in blood plasma. Did PNK treatment change the profile of microRNAs cloned from plasma? Could the authors revisit their data analysis to take a look at this?

Referee #3:

The paper by Giraldez et al describe a new method for the detection of mRNA and lncRNA in human plasma.

The paper describes a new methodology and from this point of view is really well done. The authors take in consideration a series of potential troubleshooting.

I would only add to the discussion a paragraph regarding the specific potential applications in clinic and a comment about which is the potential role of these mRNA/lncRNA in plasma.

1st Revision - authors' response

7th Apr 2019

Please see next page.

I. Referee #1 Comments:

General summary and opinion about the principle significance of the study, its questions and findings

The main significance of the study by Giraldez et al. "Phospho-sRNA-seq reveals extracellular mRNA/lncRNA fragments as potential biomarkers in human plasma" is the development of a novel method for profiling extracellular messenger RNAs (mRNA) and long noncoding RNAs (lncRNA) in plasma, for a potential use as biomarkers.

The authors added an additional step prior to library prep method, which modifies the 5' and 3' ends of small RNAs which do not have these residues naturally so that these ends will contain 5'-phosphate and 3'-OH after modification that will allow adapter ligation.

As a first proof-of-concept, the authors show that PNK treatment increases sequencing depth in sequences that lack 5'-phosphate, and sequences that lack 5'-phosphate and in addition have 3'-phosphate, from an equimolar pool. In contrast, it did not affect sequences with 5'-phosphate, most of which are microRNAs (Fig. 1). They further demonstrated that when combined with a custom, high-stringency bioinformatic pipeline, the new method, termed Phospho-sRNA-seq, reduces false positive mRNA/lncRNA fragments in plasma (Fig. 2). They further strengthened this finding by re-evaluating the top transcripts called by a standard small RNA-seq analysis pipeline with their custom pipeline (Fig. 3). In Figure 4 they demonstrate the enrichment of sense vs anti-sense transcripts in aligned reads, for both mRNAs and lncRNAs and the length distribution of the reads. It also shows the annotations for the 50 most abundant genes in the plasma of healthy patients. Figures 5 and 6 show longitudinal changes in mRNA/lncRNA fragments in the plasma of two patients undergoing bone marrow transplantation (BMT). Focus is given to bone-marrow enriched and liver-enriched transcripts and comparing the time course of changes in gene clusters with that of changes in known laboratory markers for response to the BMT (white blood cell count) and liver injury (ALT and AST plasma levels). Authors then claim that temporal changes in gene clusters correspond to changes in the well-established laboratory markers.

The study was technically well-conducted and the data seem to be solid. There is probably room / market for the new variant of RNA NGS library prep. However it is a relatively small modification and hence the authors go so demonstrate the biological value of the new method in Figure 5,6. However these clinically-related data are too preliminary. The analyses performed are not sufficient to support the value of the method and the clinically-related parts. Additional experiments should have been performed to this end, as detailed below.

Major concerns

1. Experimentally assess potential biases created by the PNK treatment. Does PNK treatment have a preference for phosphorylating/dephosphorylating of certain sequences? I am specifically concerned about the fact that PNK may act on one site in those sequences with 3'-OH (i.e. only on the 5'-OH, as there is no phosphate in the 3') and on two sites in the sequences with a 5'-OH and 3'-P, which may add to the bias.

Authors' Response: PNK treatment is known to have nucleotide sequence dependent bias, and likely also has bias dependent on whether one site or two sites are being acted upon (Lee et al. 2013). To experimentally assess the biases created by PNK treatment for recovering mRNA/lncRNA fragments in a rigorous manner, however, would require a synthetic reference set of many thousands (at the least) of synthetic RNAs of varied nucleotide sequence and 5' and 3' phosphorylation states. We feel this is beyond the scope of our current study, as our focus in this manuscript is to report that PNK treatment reveals a broad new space of RNAs that are otherwise inaccessible. We anticipate that future studies will further characterize and improve upon the methodology we have described.

We appreciate the reviewer bringing up the point of PNK bias, however, as readers should be made aware of it. We have added text to the Discussion to bring attention to the point that PNK biases exist and very likely affect the distribution of sequences recovered by the protocol described in our study. We have also added text to indicate that detailed characterization of such biases is one of the directions for further development of the approach.

2. Might PNK treatment affect adapter ligation properties? Would it not make more sense to use a library prep method that mitigates such potential bias (and not TruSeq), as the authors themselves demonstrated in their previous work (Comprehensive multi-center assessment of small RNA-seq methods for quantitative miRNA profiling, Nat. Biotech., 2018)?

Authors' Response: We are not aware that PNK treatment affects adapter ligation properties (aside from enabling the recovery of sequences that lack a 5'-phosphate and/or that possess a 3'-phosphate). We agree that using a library prep method that mitigates adapter ligation bias, such as a "4N" protocol described in the article referenced by the reviewer, could further enhance the recovery of sequences that we may be currently missing because of strong biases against them in adapter ligation. We decided to use TruSeq for the initial report of phospho-RNA-seq because TruSeq is the most commonly used small RNA library preparation protocol. Thus, it provides an accessible starting point for groups that are routinely using TruSeq to try the phospho-RNA-seq approach. We anticipate that moving forward, the phospho-RNA-seq approach will be applied to a range of library preparation protocols, including 4N protocols intended to mitigate bias. It is worth noting, however, that although 4N protocols reduce bias relative to TruSeq, substantial bias still remains. Thus, using a 4N protocol may improve the breadth of sequences captured from plasma in our study, but it would be far from "perfect" with respect to eliminating adapter ligation biases completely.

However, the reviewer's question highlights that the issue of sequence recovery bias due to library preparation method is another one that we should make readers aware of. We have added text in the Discussion to point out that the use of TruSeq as the library preparation protocol is likely producing an underestimate of the breadth of plasma mRNA/lncRNA sequences in plasma, and that using library prep methods that mitigate adapter ligation biases are likely to (i) uncover an even broader range of sequences, and (ii) provide more accurate estimates of relative abundance of different RNA fragments within a sample.

3. In order to establish the superiority over commercially available methods, additional small All of the QC measurements (i.e. % of aligned reads mapped to mRNA/lncRNA, CPM, number of

reads above a certain threshold etc.) should be assessed in more than one library prep platform RNA seq platform (not only Truseq.).

Authors' Response: We would like to clarify that our intent with this manuscript was not to establish superiority over commercially available methods, since commercially available small RNA-seq methods are not designed to recover mRNAs/lncRNAs. Rather, our intention was to develop an approach intended specifically to provide access to mRNA/lncRNA fragments in human plasma that are missed by standard commercial methods. We see the method presented as a *complement* to the commercial methods, which are best suited for recovery of microRNAs. In fact, the approach we present is inferior to commercial methods when it comes to recovery of microRNAs. We have added additional data analysis and text to make this point (see response to Reviewer 2 comment for more detail).

In addition, it is worth noting that many other commercial small RNA-seq library preparation methods (aside from TruSeq) are also designed to capture microRNAs based on presence of a 5'-P and 3'-OH. Thus, given the premise and results of our manuscript, it is unlikely that the overall conclusion that PNK increases recovery of mRNA/lncRNA fragments will be changed by using other commonly used downstream library preparation methods.

4. In Figure 4C, it seems that about half of the 50 most highly abundant genes greatly vary in length (looking at the dots to the right of the boxplots). What is the reason for that?

Authors' Response: We also found this observation interesting. We don't have a definitive explanation for this currently. However different RNA molecules in plasma could have differing stability depending on different carriers or protective mechanisms against nucleases. Plasma ribonuclease activity may also vary between individuals. Based on these possibilities, it is perhaps not surprising to see length variation in these plots, which show just the average length for the most abundant fragments in 5 individuals. Our comments on this remain speculative, however, until future data provides a definitive answer.

5. The claim that specific exRNA transcript fragments might provide access to a circulating transcriptome with potential for biomarker development is true in general. However the data and analysis provide is not mature enough to support the specific claim. Thus, for data presented in Fig. 5C and 6C, D:

- a. The figures only show a general trend, but there is no report on correlation between levels of mRNAs/lncRNAs to levels of ALT/AST and numbers of white blood cells.
- b. The levels of the genes in P04 and P07 cluster do not seem to change longitudinally to the same extent as ALT/AST and white blood cell, and have more moderate peaks. Therefore, it does not seem suitable for tracking time-dependent changes.

Authors' Response to Reviewer Comments 5a and 5b: We would like to clarify that our intent in this manuscript is only to claim that exRNA fragments might provide access to a circulating transcriptome with potential for biomarker development *in general* (as the reviewer notes), and not to claim that we have identified specific biomarkers for management of hematopoietic stem cell transplant (HSCT) patients. This intention motivated us to include the adjective "potential" as a modifier of the word "biomarkers" in the manuscript Title.

The main point of showing the data in Figures 5 and 6 was to demonstrate that gene sets related to bone marrow and liver, which are expected to change through the course of HSCT and liver toxicity, respectively, do show dynamic changes in a pattern expected based on known biology. Thus, the white blood cell (WBC) counts and AST/ALT levels are intended for validation that the biological processes of bone marrow suppression followed by reconstitution, and liver toxicity, did indeed occur in these patients. The dynamic changes in plasma mRNA/lncRNA fragments corresponding to bone marrow and liver enriched genes shows that the fragments we are seeing are not an artifact, but rather inform about true, expected biological changes, based on changes in organ-specific RNA sets observable in plasma. We are not intending to claim that these gene signatures would be suitable for tracking time-dependent changes in BMT patients for potential clinical use.

Thus, our intention with the lab values was not to use them as a ground truth by which to assess the accuracy mRNA/lncRNA fragment profiling data. In fact, on biological grounds we don't expect a perfect correlation of extracellular RNA fragment gene signature quantitative values with ALT/AST and WBC counts. The mechanisms that cause release of ALT/AST protein into the circulation and its half-life there, for example, could be quite different from those governing release and plasma half-life of extracellular RNA fragments. The same may be true of WBC counts relative to extracellular RNA. Such differences can be expected to affect the degree of correlation between the different types of data. Again, our purpose with Figures 5 and 6 was simply to provide proof-of-concept data to support the general claim that transcriptomic signatures of extracellular RNA fragments in plasma do exist and can dynamically reflect changing biological processes. Based on our data, we propose that this approach is worth further exploration in future studies, to identify and validate specific biomarkers for clinical needs in a variety of disease settings.

c. The predictive power of lncRNAs and mRNAs should be determined in a larger and independent set of patient samples and healthy controls (replication study) and compared to predictive power of well-known and well-established biomarker, in order to support the main claims of the authors.

Authors' Response: We agree that future studies will be needed to demonstrate the full potential of mRNA and lncRNA fragments as disease biomarkers in specific clinical settings. As explained above, our intention in this study was to demonstrate that these fragments exist in plasma, and to provide proof-of-concept that they display characteristics that indicate potential for clinical biomarker development. Hence in our title we refer to these as "potential biomarkers". We envision that publication of our manuscript will enable diverse labs to use the extracellular RNA fragment sequencing approach we describe to discover plasma mRNA/lncRNA fragment biomarkers for a range of clinical needs. In such studies, using independent sets of specimens (e.g., independent training set and test set) as mentioned by the reviewer, will be important for validating biomarkers being developed for clinical use.

Minor concerns that should be addressed

1. The author state that miRNAs are "exceptionally stable in plasma" (p.2), and further state that the stability of mRNAs and lncRNAs in plasma is questionable ("mRNAs and lncRNAs, if truly

stable in blood plasma at all..." , p. 3). If so, then explain why looking at mRNAs and lncRNA as markers is of value over measuring miRNAs?

Authors' Response: The phrases referred to are from the Introduction. They are used to provide context for why we thought to search for fragments of mRNAs, given that we expected long fragments would not remain fully intact in the RNase-rich environment. The fact that we find mRNA fragments, and that many of them are detected in multiple samples and from different individuals, provides evidence that they are stable (i.e., at least stable enough to be recovered in our sequencing studies).

Also, we are not claiming that looking at mRNA and lncRNA fragments is universally superior to microRNAs. Rather, we believe that mRNA and lncRNA fragments provide additional, complementary information and that the entire plasma transcriptome is worthy of exploration.

2. Explain why the first experiments in equimolar synthetic pool was performed with the NextSeq platform and subsequent experiments in plasma were performed in HiSeq.

Authors' Response: The synthetic pool sequencing was done on a NextSeq platform because synthetic samples exhibit a lesser degree of complexity than biological samples and, therefore, require less sequencing depth. Plasma samples were sequenced on HiSeq to obtain greater sequencing depth.

3. Why was the input material for library prep determined by volume (5l) and not RNA amount in ng?

Authors' Response: The RNA yield from plasma RNA extraction is usually too low to quantify concentrations accurately. Standard practice in the field of extracellular RNA is to use RNA extracted from a standardized volume of plasma, rather than a standard mass of RNA as input into library preparation. This is the same practice employed by the Extracellular RNA Communication Consortium in our prior study referenced by the Reviewer (Giraldez, Spengler et al, Nature Biotech, 2018).

4. In figure 1B, show PNK- box plot at the left hand side, so that it is clear that PNK treatment increases the CPM for sequences with 5'-OH and 3' P/OH.

Authors' Response: The suggested change has been made.

5. In Figure 2E, only absolute numbers of uniquely mapped and multi-mapped reads are presented. It seems that both are increased by PNK. Thus, is there a change in the % of uniquely and multi-mapped reads as well?

Authors' Response: The % of uniquely-mapped and multi-mapped reads is roughly comparable between PNK+ (80.4% uniquely mapped; 19.6% multi-mapped) and PNK- (85.7% uniquely mapped; 14.3% multi-mapped) samples.

6. In p. 9, the authors refer to "Figure 4D and Appendix Table S3" but it might be referring to Figure 4C that shows the top 50 expressed genes.

Authors' Response: We thank the reviewer for pointing out this error. The figure reference has been corrected.

Summary: The manuscript falls short of the required novelty and its overall contribution to the field is premature and hence below what I wish to see at the EMBO Journal.

Authors' Response: Although the method we present here is a simple modification of currently used small RNA-seq protocols, it is an important one because it provides access to a substantial space of mRNA/lncRNA fragments in human plasma, which are largely missed by standard protocols. In addition, it is not only the experimental modification but also the high-stringency bioinformatic pipeline that we created, which together enable access to these fragments. Showing that dynamic patterns of plasma mRNAs/lncRNAs reflect pathophysiological processes provides proof-of-concept for applying the approach for biomarker discovery in different settings. We believe that publishing this novel approach will facilitate the discovery of new RNA-based biomarkers for a range of medical conditions and will help move the emerging field of liquid biopsy forward.

II. Referee #2 Comments:

This paper reports an improved method for cDNA cloning, sequencing, and analysis of RNA transcript sequences in circulation in human blood plasma. Methods for sequence analysis of circulating microRNAs have been well established, but previously no good methods were available for sequencing mRNA or lncRNA transcripts in circulation, apparently for the reason that, unlike microRNAs, such degradation products of longer transcripts would be expected to lack 5' phosphate, and often contain a 3' phosphate. The authors devised a method that employs T4 polynucleotide kinase treatment of RNA extracted from plasma, followed by standard ligation-based cDNA cloning, and they show that their method substantially enhances the yield of sequence reads from the sense strands of mRNAs and lncRNAs. They employ their method in the context of longitudinal studies of patients undergoing bone marrow transplants, with the results showing apparent dynamic signatures of bone marrow and liver transcripts. The authors also demonstrate the importance of employing stringent computational filters to mitigate the false-positive effects caused by highly abundant and repetitive sequence reads. This is a valuable new method, well documented by convincing data. Publication in EMBO Journal is encouraged. There are only two minor critiques:

1) Figure 6C and 6D do not seem to be specifically cited in the text.

Authors' Response: We thank the reviewer for pointing out this omission, and have modified the text to reference these figures.

2) Please provide complete reaction conditions for the T4 PNK treatment.

Authors' Response: We have modified the methods to include the complete reaction conditions.

3) It seems like the authors may have missed an opportunity to address whether there may be a population of unphosphorylated microRNA in blood plasma. Did PNK treatment change the profile of microRNAs cloned from plasma? Could the authors revisit their data analysis to take a look at this?

Authors' Response: We have revisited our data analysis to study the impact of PNK treatment on recovery of microRNAs. We find that PNK treatment markedly reduces the microRNA footprint in these libraries, presumably by competition from non-microRNA fragments that are made accessible in library preparation by PNK treatment. The difference is substantial enough that traditional library preparation methods seem preferable for microRNA profiling. We have modified the manuscript text to include this data and make readers aware of this important point (see **Figure EV1** in the Expanded View section).

Also at the reviewer's suggestion, we analyzed the microRNA alignments and did find examples of microRNAs specifically detected with PNK treatment only, despite the lower read coverage of microRNAs with PNK treatment. Closer examination of these microRNAs revealed that most were reads from transposable elements and/or microRNAs which have since been removed from miRBase. However, there were at least 8 *bona fide* microRNAs with no reads detected in any of the healthy individual samples without PNK treatment, but which were detected in all five of the same individuals after PNK treatment. We have modified the text to describe this analysis and added the results to the supplementary information in the Expanded View section (**Figure EV1**).

III. Referee #3 Comments:

The paper by Giraldez et al describe a new method for the detection of mRNA and lncRNA in human plasma.

The paper describes a new methodology and from this point of view is really well done. The authors take in consideration a series of potential troubleshooting.

I would only add to the discussion a paragraph regarding the specific potential applications in clinic and a comment about which is the potential role of these mRNA/lncRNA in plasma.

Authors' Response: We thank the reviewer for the complimentary assessment. We have expanded our discussion to include specific potential applications in the clinic, as well as the potential functional role of these extracellular mRNAs/lncRNAs in plasma.

Additional Changes Made to the Manuscript (Note that all changes are indicated by *Red text* in the manuscript document):

- Added middle initial "J." for author A.G., so it reads "Annika J. Goicochea".
- Wording changes for grammar, style, additional technical detail and clarity in the manuscript text.

- We noticed that Figure 6 was being called in the text before Figure 5B and Figure C. We re-ordered some sentences in the Results section so that the Figure panels are all called in correct sequence from the text.
- We added some additional references, including for the EBSeq-HMM R package (Leng et al. 2015) and for sRNAAnalyzer (Wu et al. 2017).
- In the Computational Methods section, we expanded on and corrected some details in the Synthetic Pool Library Analysis paragraph, which had been missed in the original version due to an oversight.
- We replaced “bone marrow transplant (BMT)” was with “hematopoietic stem cell transplant (HSCT)” throughout the manuscript, as this is a more contemporary terminology for this treatment.
- We slightly changed the terminology from “phospho-small RNA-seq” or “phospho-sRNA-seq”, to “phospho-RNA-seq” for a couple of reasons: (i) the former terminology could be misleading, since the approach is not restricted only to finding “small RNAs”, since we observed a range of insert sizes and the definition of “small RNAs” is arbitrary, and (ii) the shorter version is more practical for common use as a shorthand for the method.
- In order to reduce the chance of confusion between the abbreviation ‘mRNA’ and ‘miRNA’, we removed the use of the abbreviation ‘miRNA’ and spelled out “microRNA” throughout the transcript.

2nd Editorial Decision

11th Apr 2019

Thanks for submitting your revised manuscript to the EMBO Journal. I have now had a chance to take a look at it and I appreciate the introduced changes. I am therefore very pleased to let you know that we will accept the manuscript for publication here.

Before I can send you the formal acceptance letter there are just some editorial things we have to sort out.

2nd Revision - authors' response

14th Apr 2019

The authors performed all requested editorial changes.

Corresponding Author Name: Muneesh Tewari

Journal Submitted to: The EMBO Journal

Manuscript Number: EMBOJ-2019-101695